# Recent Advances in Personal Glucose Meter-Based Biosensors for Food Safety Hazard Detection

**DOI:** 10.3390/foods12213947

**Published:** 2023-10-29

**Authors:** Su Wang, Huixian Huang, Xin Wang, Ziqi Zhou, Yunbo Luo, Kunlun Huang, Nan Cheng

**Affiliations:** 1Beijing Laboratory for Food Quality and Safety, College of Food Science and Nutritional Engineering, China Agricultural University, Beijing 100083, China; chicagow777@cau.edu.cn (S.W.); davidwx@cau.edu.cn (X.W.); zhouziqi214@163.com (Z.Z.); lyb@cau.edu.cn (Y.L.); hkl009@163.com (K.H.); 2College of Environmental and Food Engineering, Liuzhou Vocational and Technical College, Liuzhou 545000, China; huanghx6756@163.com; 3Key Laboratory of Safety Assessment of Genetically Modified Organism (Food Safety), Ministry of Agriculture, Beijing 100083, China

**Keywords:** personal glucose meter, food safety hazards, point-of-care testing, biosensors

## Abstract

Food safety has emerged as a significant concern for global public health and sustainable development. The development of analytical tools capable of rapidly, conveniently, and sensitively detecting food safety hazards is imperative. Over the past few decades, personal glucose meters (PGMs), characterized by their rapid response, low cost, and high degree of commercialization, have served as portable signal output devices extensively utilized in the construction of biosensors. This paper provides a comprehensive overview of the mechanism underlying the construction of PGM-based biosensors, which consists of three fundamental components: recognition, signal transduction, and signal output. It also detailedly enumerates available recognition and signal transduction elements, and their modes of integration. Then, a multitude of instances is examined to present the latest advancements in the application of PGMs in food safety detection, including targets such as pathogenic bacteria, mycotoxins, agricultural and veterinary drug residues, heavy metal ions, and illegal additives. Finally, the challenges and prospects of PGM-based biosensors are highlighted, aiming to offer valuable references for the iterative refinement of detection techniques and provide a comprehensive framework and inspiration for further investigations.

## 1. Introduction

With the continuous advancement of technology, higher demands have been placed on food quality and safety. Food safety risk factors such as pathogenic microorganisms, fungal toxins, agrochemical residues, heavy metal ions, and unauthorized additives not only jeopardize consumer health but also have adverse impacts on societal stability. Diverse techniques have been applied to the detection of food safety hazards, including high-performance liquid chromatography [1], liquid chromatography-mass spectrometry [2], gas chromatography, gas chromatography-mass spectrometry [3], and nuclear magnetic resonance [4]. While these methods offer elevated sensitivity and precision, their intrinsic limitations, such as intricate sample preparation, expensive instrumentation, and the requirement for skilled operators, curtail their application within the realm of food analysis.

Biosensors, characterized by their attributes of cost-effectiveness, portability, ease of fabrication, and operational simplicity, have exhibited substantial potential within the realm of food analysis [5]. Biosensors amalgamate biological recognition elements with signal transduction components to transduce biological reactions into quantifiable physical or chemical signals, thereby constituting analytical devices for the detection of target compounds [6]. Depending on their different output signals, biosensors can be categorized into diverse types such as electrochemical and optical sensors [7]. Among them, optical biosensors can be categorized into different types, such as fluorescence, colorimetric, surface-enhanced Raman scattering (SERS), surface plasmon resonance (SPR), etc. [8]; the probability of large instrument equipment serving as optical signal outputs remains significant. And the electrochemical methods for detecting specific components in food are currently the most effective approach due to their cost-effectiveness, rapid signal response, high sensitivity, and user-friendliness [9]. By harnessing interactions with antibodies, aptamers, DNA enzymes, etc., an array of rapid food safety detection methods has been developed, including enzyme-linked immunosorbent assay (ELISA) kits, lateral flow assay (LFA), and others [10]. Nonetheless, the majority of these methods are restricted to qualitative or semi-quantitative analysis, revealing the imperative to devise more precise techniques for the quantitative assessment of food safety hazards.

Ever since their inception, personal glucose meters (PGMs) have emerged as one of the most widely utilized devices for monitoring blood glucose levels in diabetic patients, attributed to their portability, cost-effectiveness, ready accessibility, and rapid detection capabilities [11]. PGMs as intelligent devices are composed of glucose meters and test strips. The glucose meter is employed to ascertain glucose concentration based on variations in current or voltage, while the test strip comprises an enzymatic electrode housing dehydrated forms of glucose oxidase or dehydrogenase. These enzymatic elements facilitate enzyme-mediated reactions with glucose in blood, inducing alterations in current or voltage that closely correlate with glucose concentration [12]. Initially, PGMs were confined to glucose detection, thereby considerably limiting their application scope within other domains of rapid analysis. However, in 2011, Lu et al. [13] pioneered the linkage of glucose detection with the evaluation of diverse targets through the employment of functional DNA-coupled glucose invertase. The researchers quantified non-sugar substances within samples based on glucose concentrations, thereby transcending the conventional targeting limitations of PGMs. Subsequent to this pivotal breakthrough, PGMs have undergone a rapid and robust evolution towards the detection of non-sugar substances.

This paper presents a comprehensive overview of the mechanisms underpinning the construction of PGM-based biosensors, comprising three fundamental components: recognition, signal transduction, and signal output. It also meticulously enumerates the various recognition and signal transduction components, and outlines their strategies of integration. Then, a multitude of instances is examined to present the latest advancements in the application of PGMs in food safety detection, including targets such as pathogenic bacteria, mycotoxins, agricultural and veterinary drug residues, heavy metal ions, and illegal additives. The strengths and limitations of different research approach are analyzed. Finally, the challenges and prospects of PGM-based biosensors are highlighted, aiming to offer valuable references for the iterative refinement of detection techniques and provide a comprehensive framework and inspiration for further investigations.

## 2. Construction of PGM-Based Biosensors

The construction of PGM-based biosensors primarily encompasses three fundamental components: recognition, signal transduction, and signal output. The specific construction mechanism is illustrated in Figure 1. First is the recognition component, which achieves a specific recognition of the target analyte through molecular interactions. Commonly employed recognition elements include antibodies, aptamers, DNA enzymes (DNAzymes), molecularly imprinted polymers (MIPs), and antimicrobial peptides (AMPs). Subsequently, the signal transduction element establishes the connection between the recognition element and the signal output element, converting molecular recognition events into quantifiable or measurable signals. Frequently utilized signal transduction elements include glucose invertase and glucose oxidases. Among these, invertase hydrolyzes sucrose to produce glucose and fructose, while glucose oxidase acts on glucose to generate gluconic acid and hydrogen peroxide. The coupling methods between recognition elements and signal transduction elements typically involve the construction of controlled release systems for signal transduction elements, functionalizing oligonucleotides or antibodies with invertase labels, exploiting the bridging effect of nanomaterials, and synthesizing organic–inorganic nanoclusters. Lastly, the signal output element, represented by PGMs, amplifies or outputs measurable electrochemical signals, ultimately translating the displayed numerical values on PGMs into the concentration of the analyte under investigation [14].

### 2.1. Recognition Elements

#### 2.1.1. Antibodies

Antibodies, also known as immunoglobulins (Igs), are immunoglobulin proteins generated by the adaptive immune system that exhibit high specificity for particular pathogens [15]. In 1959, Yalow et al. [16] pioneered the use of antibodies as sensing elements in immune analysis for insulin. Subsequently, antibodies have been extensively researched and applied as ideal biorecognition elements. Antibody-based biosensors have profoundly revolutionized the detection methodologies for biomarkers, food contaminants, illicit drugs, and more, and continue to evolve robustly to the present day. Dou et al. [17] employed antibodies as recognition elements and PGMs as quantitative instruments to develop a portable and efficient method for the quantification of *Escherichia coli* (*E. coli*), specifically *E. coli* O157:H7. The method holds potential for adaptation to other bacterial species by modifying the employed antibodies. However, this approach utilized LFA as a sensing platform, and the cutting of lateral flow strip test lines to integrate with PGMs remains necessary in this method. Considering approaches to effectively address the challenge of seamlessly integrating test strips and PGMs, as well as improving the alignment between LFA and PGMs, is necessary.

#### 2.1.2. Aptamers

The term “aptamer” originates from the Latin word “aptus” (fit) and the Greek word “meros” (part) [18]. The first aptamer was reported by Ellington et al., in 1990 [19]. Aptamers belong to the family of functional nucleic acids and consist of single-stranded DNA or RNA oligonucleotides, typically ranging in length from 3 to 80 nt. They achieve high affinity and specific recognition for targets through hydrogen bonding, electrostatic interactions, and hydrophobic interactions [20]. Aptamers exhibit specific binding to a diverse array of target molecules, including nucleic acids, proteins, small molecules such as metal ions, etc. Their small molecular weight, ease of synthesis, modification, low cost, and non-immunogenicity have collectively propelled their application expansion, positioning them as a substitute for antibodies [21]. This versatility has led to their widespread adoption in various domains of food analysis. Recently, Gu et al. [22] harnessed aptamers to construct a high-affinity PGM-based biosensor of melamine. This portable, rapid, and quantitative sensor was developed for the efficient detection of melamine in milk. Similarly, Yang et al. [23] developed a magnetic aptamer-PGM biosensor for the detection of *Staphylococcus aureus* in food samples. The method demonstrated high sensitivity and specificity, with results consistent with those obtained using the plate counting method. Notably, the integration of aptamers with the PGM sensing platform substantially reduced the required detection time.

#### 2.1.3. DNAzymes

In the 1990s, Breaker et al. [24] pioneered the discovery of catalytically active DNA sequences, known as DNA enzymes or DNAzymes, using a combination of in vitro selection known as systematic evolution of ligands by exponential amplification (SELEX). DNAzymes are composed of enzyme strands and substrate strands, with the substrate strands containing single-stranded RNA junctions (such as rA) that serve as cleavage sites [25]. When enzyme strands bind to their target metal ions (such as Pb^2+^, Ca^2+^, Mg^2+^), they exhibit enhanced catalytic activity, enabling the recognition and cleavage of substrate strands containing rA. Compared to protein enzymes, these metal-dependent DNAzymes offer high selectivity and sensitivity. They are stable, easily stored, and amenable to modifications [26]. These attributes make DNAzyme-based biosensor platforms for heavy metal ion detection especially attractive. Tang et al. [27] harnessed Pb^2+^-specific DNAzymes as recognition elements to develop a portable quantitative biosensor based on PGMs. The method involves encapsulating glucose-loaded mesoporous silica nanoparticles (MSN) with Pb^2+^-specific DNAzymes. In the presence of the target Pb^2+^, the substrate strand is cleaved, resulting in the release of glucose from the MSN and subsequently generating a quantifiable signal on PGMs. This approach is rapid and field-deployable, offering versatility through the interchange of specific DNAzymes for the universal detection of other heavy metal ions.

#### 2.1.4. Molecularly Imprinted Polymers

The history of MIPs can be traced back to 1973 [28], with a series of papers published by the Wulff research group titled “Enzyme-Analog Built Polymers”. The first paper that explicitly reported an “imprinted polymer” was co-authored by K. Mosbach and B. Sellergren in 1984 [29]. In 1993, K.Mosbach et al. [30] published a study in the journal Nature detailing the preparation of theophylline molecularly imprinted polymers (MIPs) using a non-covalent approach. This work garnered significant attention from researchers, sparking widespread interest and investigation into the field of molecular imprinting technology, consequently fostering its robust development. MIPs are crosslinked polymers possessing internal cavities with predetermined sizes, shapes, and specific functional group arrangements [31]. MIPs exhibit commendable attributes such as structural predictability, recognition specificity, mechanical, thermal, and chemical stability, high affinity, low cost, and ease of synthesis. These advantages position MIPs as notably superior to natural recognition materials and facilitate their wide-ranging application in sensing domains [32]. However, certain limitations persist, including irregular morphology, deeply embedded pores, incomplete template removal, template leakage, non-uniform distribution of binding sites, and sluggish rebinding kinetics [33], thereby imposing practical constraints on their application. To further enhance MIP performance, contemporary research has predominantly embraced core–shell structures, wherein MIP shells encapsulate nanostructured cores. For instance, Li et al. [34] developed a classic sandwich method based on PGMs for the rapid detection of chloramphenicol (CAP) in food. In this approach, the recognition elements encompass magnetic molecularly imprinted probes (m-MIP) and the β-cyclodextrin/invertase (EV-Au-β-CD/INT) complex. Specifically, m-MIP probes are employed for capturing the 2,2-dichloroacetamide fragment of CAP, while the EV-Au-β-CD/INT complex is utilized to capture the residual nitrobenzene fragment within CAP. Following a magnetic separation step, the resultant m-MIP/CAP/EV-Au-β-CD/INT complex catalyzes the hydrolysis of sucrose into glucose, thereby producing a quantifiable signal detectable by PGMs. This sensing platform exhibited outstanding specificity and efficiency, thereby demonstrating the excellence of this approach.

#### 2.1.5. Antimicrobial Peptides

Antimicrobial peptides (AMPs), also known as antimicrobial proteins, are short peptide fragments consisting of 6 to 50 positively charged amino acid residues and a significant proportion of hydrophobic residues. These peptides possess amphiphilic properties and exhibit a strong affinity for negatively charged bacterial membranes, allowing them to efficiently target bacterial surfaces [35]. In the 1980s, the research group led by Swedish scientist Boman induced the production of antimicrobial peptide-like substances in the cecropia moth (Hyalophora cecropia) following Bacillus cereus induction, leading to the discovery of the first AMPs, cecropins [36]. Compared to antibodies and aptamers, AMPs demonstrate greater stability. They exhibit high affinity and specificity towards various analytes such as proteins, nucleic acids, metal ions, and bacterial cells. This efficacy as recognition elements has prompted widespread attention for their use in developing sensitive and convenient biosensors [37]. Bai et al. [38] utilized AMPs as recognition elements to establish a sandwich immunoassay for the rapid detection of *E. coli* O157:H7. In this method, they synthesized AMP-modified nanocomposites. Using a one-step procedure, they employed magainins I and iron oxide nanoparticles embedded with copper phosphate as recognition elements to capture *E. coli* O157:H7. Subsequently, they employed cecropin P1 and enzyme-synthesized calcium phosphate nanoparticles as signal labels. The enzyme could hydrolyze sucrose into glucose, thereby converting *E. coli* O157:H7 levels to glucose levels, quantified through PGMs. Under optimal conditions, the concentration of *E. coli* O157:H7 could be determined within a linear range of 10 to 10^7^ CFU/mL, with a detection limit of 10 CFU/mL. This method was successfully applied to the determination of *E. coli* O157:H7 in milk samples.

#### 2.1.6. Alternative Approaches

The inherent characteristics of the target can also serve as input signals for sensing systems. For instance, Chavali et al. [39] leveraged the intrinsic property of *E. coli* to produce β-galactosidase, enabling the rapid quantitative detection of *E. coli* through the construction of a PGM sensing platform. In this method, lactose was employed as the substrate. When the target *E. coli* was present, the generated β-galactosidase cleaved the lactose β-1,4-glycosidic bond, resulting in glucose production, which was subsequently quantified by PGMs without the need for complex operations. Five years later, Tang et al. [40] harnessed the inhibitory ability of organophosphates on the catalytic activity of acetylcholinesterase (AChE) to develop a PGM-based sensor for the rapid quantitative detection of paraoxon in food. During the detection process, AChE catalyzed the hydrolysis of acetylthiocholine chloride to produce thiocholine. When the target paraoxon was present, its catalytic activity was inhibited, leading to a reduction in thiocholine content. Consequently, the reduction of [Fe(CN)6]^3−^ to [Fe(CN)6]^4−^ was impeded, restricting electron transfer. Due to the detection of thiocholine in the same manner as the detection of glucose in blood, the above reactions can be measured by PGMs. This method achieved a detection limit of 5 µg/L for paraoxon. Notably, its advantage lies in the direct impact of [Fe(CN)6]^3−^, which differs from the intermediate invertase utilized in many methods. This approach eliminates the need for complex modification and synthesis processes, rendering it a more meaningful strategy.

### 2.2. Signal Transduction Elements

Given that PGMs are designed to detect only glucose, the dynamic range for monitoring blood glucose levels is typically 0.6 to 33 mM [41]. However, when it comes to the detection of other target substances related to food, medical, or environmental analysis, the required detection concentrations often fall within the nmol or μmol range. Consequently, to establish a PGM-based biosensor, an efficient signal amplification process is necessary to enhance detection efficiency by approximately 10^6^ times [13].

Researchers have discovered that the enzyme β-D-fructofuranosidase (invertase) can fulfill this signal amplification function, thereby offering the potential to serve as a signal transduction element for PGM biosensors. Firstly, invertase can catalyze the hydrolysis of sucrose into fructose and glucose. Since the substrate sucrose is completely inert in the PGM test strip, while the hydrolysis product glucose can be monitored by PGMs, a direct relationship can be established between the concentration of invertase and that of glucose. This relationship can be utilized to correlate the concentrations of glucose with other target substances, facilitating comprehensive detection within the realm of food safety. Secondly, invertase exhibits high enzymatic activity. Even at the nmol level, it can effectively convert mmol concentrations of sucrose to glucose under environmental conditions. Consequently, invertase emerges as an ideal candidate medium for converting other target substances into glucose [42]. Additionally, glucose oxidase and glucoamylase can perform functions similar to those of invertase, thus serving as alternative signal transduction components [43].

### 2.3. Integration Strategies of Recognition and Signal Transduction Elements

The coupling of recognition elements with signal transduction components represents a pivotal aspect in the design and functionality of biosensing systems. This strategic amalgamation enables the conversion of molecular interactions between target analytes and recognition elements into quantifiable signals that can be detected and measured, ultimately facilitating the determination of analyte concentrations. In biosensing configurations based on PGMs, the integration strategies of recognition and signal transduction mainly include the following: establishing controlled release systems for signal transduction elements, functionalizing the recognition element with signal transduction elements labels, employing nanoparticles as bridging agents, and synthesizing organic–inorganic nanoclusters as efficient carriers for both the recognition and signal transduction elements.

#### 2.3.1. Target-Responsive Controlled Release Systems

Controlled release systems in PGM sensors primarily involve transduction elements encapsulating materials such as mesoporous silica nanoparticles (MSNs), liposomes, hydrogels [44], et al., and gating agents, including polymers [45], nanoparticles [46], supramolecular assemblies [47], biomolecules [48], et al. Recently, aptamer-gated MSNs have gained considerable attention as a controllable nanocontainer for glucose release, owing to aptamers’ notable features of high affinity, specificity, thermal and chemical stability, and ease of acquisition. Wang et al. [49] devised a “dual-gated” approach for instantaneous detection of aflatoxin B1 (AFB1) using the PGM readout system. The “dual gate” concept entailed the sequential encapsulation of aptamers and dopamine on amine-modified MSN composites. Under acidic conditions (pH 5.5), the introduction of AFB1 instigated dopamine’s self-degradation, subsequently causing it to bind with the aptamer and trigger the opening of the “dual gates.” Consequently, encapsulated glucose was released, producing a measurable PGM signal. In a similar vein, Yan et al. [50] employed aptamer-crosslinked hydrogels to encapsulate glucose oxidase, creating a PGM-based biosensor for rapid cocaine detection in food samples. In the presence of cocaine, the preferential recognition between cocaine and its aptamer prompted hydrogel degradation and the release of glucose oxidase. Subsequently, glucose oxidase hydrolyzed linear starch in the substrate, effectively transforming cocaine detection into glucose generation for subsequent PGM readout. This method obviates the need for intricate chemical enzyme modifications, thereby optimizing enzyme activity and achieving a low detection limit of 3.8 μM, comparable to commercial cocaine test kits.

#### 2.3.2. Functionalization of Recognition Elements

Another mature strategy for PGM sensors involves the functionalization of recognition elements based on enzymes, such as invertase [51]. Commonly used single-stranded nucleic acid probes, such as aptamers, DNAzymes, or complementary sequences to the target, can be chemically cross-linked to invertase using thiol-SMCC crosslinkers. For the coupling of antibodies with invertase, avidin–biotin interactions are often employed as intermediates to create biotinylated antibodies and biotinylated invertase [52]. This relationship between non-glucose targets, specific recognition elements, and invertase allows the PGM-based biosensors to possess both recognition and quantification capabilities for non-glucose targets. Lu et al. [53] coupled aptamer complementary sequences with invertase (DNA-INV) and aptamer-modified magnetic beads, placing them onto the binding pad of LFA. In the presence of the target cocaine, the formation of the cocaine–aptamer complex leads to the release of DNA-INV, which converts sucrose added to the reaction pad into glucose, enabling quantitative detection by PGMs. The detection limit of this method is slightly reduced compared to liquid-phase reactions. This might be attributed to the loss of released DNA-INV before entering the LFA reaction pad. Therefore, how to minimize the loss of invertase during the chromatography process to enhance sensitivity becomes an area worth investigating.

#### 2.3.3. Nanomaterials as Bridging Agents

Nanomaterials exhibit notable attributes including high surface area, enhanced reactivity, and remarkable miniaturization, rendering them well suited to the multifunctionality, miniaturization, and rapidity requisites of modern biosensors. Notably, gold nanoparticles (AuNPs) [54] and magnetic nanoparticles (MNPs) [55] have garnered substantial attention in biosensing applications. The inherent characteristics of AuNPs and MNPs, including their abundant active surface sites, robust adsorption capacity, and elevated electron density, empower them to adsorb various biomolecules such as nucleic acids and proteins rapidly and stably without impairing their bioactivity. This is further complemented by their outstanding biocompatibility, rendering them versatile biomolecule carriers. In the context of addressing issues such as *Salmonella* contamination and infection, Joo et al. [56] proposed a convenient strategy for detecting *Salmonella* in milk through the utilization of antibody-functionalized MNPs. In this approach, *Salmonella* is initially captured from milk samples by employing monoclonal antibody-functionalized magnetic nanoparticle clusters (MNCs). Subsequently, polyclonal antibody-functionalized invertases are introduced, effectively linking the MNC–*Salmonella* complex with bounded invertases. This innovative methodology achieves a remarkable detection sensitivity of 10 CFU/mL, showcasing pronounced selectivity for effectively isolating *Salmonella* from milk. Importantly, both detection limits and assay durations surpass those of previous reports. The utilization of AuNPs and MNPs in biosensing highlights their versatile capabilities in effectively addressing diverse biosensing challenges.

#### 2.3.4. Synthesis of Organic–Inorganic Nanoflowers

The organic–inorganic hybrid nanoflowers can combine the functionalities of enzymes and inorganic materials, bridging signal input and conversion elements. In 2012, Ge et al. [57] first reported a method for synthesizing organic–inorganic hybrid nanoflowers. They employed copper ions as the inorganic component and various proteins as the organic component to form complexes. These complexes served as nucleation sites for the growth of copper phosphate primary crystals. The interaction between proteins and copper ions led to the development of micrometer-sized particles with nanoscale features resembling flower petals. When enzymes were utilized as the protein component of the hybrid nanoflowers, the high surface area of the nanoflowers and the enzyme’s substantial loading capacity contributed to enhanced enzyme activity and stability compared to free enzymes. This integration facilitated the amalgamation of biological recognition and signal amplification functionalities, offering significant prospects in the field of biosensing. In 2016, Lin et al. [58] employed a one-pot biologically induced synthesis to prepare concanavalin A (Con A)–invertase–calcium hydrogen phosphate (CaHPO_4_) nanoflowers. In this context, Con A exhibited strong affinity to *E. coli* surface O antigen, while CaHPO_4_ provided biocompatible sites for Con A and invertase modification, thereby enhancing invertase activity. As a result, the synthesized Con A–invertase–CaHPO_4_ nanoflowers not only facilitated the recognition of *E. coli* O157:H7 but also efficiently converted sucrose added to the system into glucose, yielding quantifiable PGM signals. The method demonstrated a detection sensitivity of 10^1^ CFU/mL, thus representing a rapid and effective approach for *E. coli* O157:H7 detection.

## 3. Application of PGMs in Food Safety Hazard Detection

Due to their unique advantages, such as portability, short detection time, and quantifiability, PGM-based biosensors have been extensively researched and gradually applied in the field of food safety hazard detection. They are being employed for the detection of various contaminants in food, including pathogenic bacteria, mycotoxins, agricultural and veterinary drug residues, heavy metal ions, illegal additives, and more (refer to Table 1).

### 3.1. Detection of Foodborne Pathogens

Foodborne pathogens encompass a diverse array of microorganisms, including species such as *Vibrio*, *Listeria* monocytogenes, *Salmonella*, *Staphylococcus aureus*, *Clostridium perfringens*, *E. coli* O157:H7, and various Shiga toxin-producing *E. coli* strains, among others [75]. In recent years, the significance of foodborne pathogens in causing foodborne illnesses and poisoning incidents, thereby posing substantial threats to public health and safety, has underscored the pressing need for swift methods to detect these pathogens in food samples.

As illustrated in Figure 2A, Huang et al. [59] combined a dual-layer capillary high-gradient immunomagnetic separation approach with a plasmonic-gold-modified nanoflower signal amplification scheme to develop a PGM biosensor for the sensitive and rapid detection of *E. coli* O157:H7. This methodology involved the initial coupling of biotinylated polyclonal antibodies against *E. coli* O157:H7 to magnetite nanoparticles (MNBs) modified with streptavidin, yielding immunomagnetic nanoparticles (IMNBs) as the recognition elements for capturing target bacteria. Subsequently, monoclonal antibodies and glucose oxidase were conjugated onto the *E. coli* O157:H7 cells, resulting in the formation of sandwich immunoassay complexes of MNB–bacteria immunocomplexes (INCs) within the capillary. The injection of sucrose into the capillary facilitated enzymatic conversion by glucose oxidase, leading to glucose production and subsequently generating quantitative PGM signals. The limit of detection achieved for *E. coli* O157:H7 was 79 CFU/mL, with a dynamic range spanning from 10^2^ to 10^7^ CFU/mL. The recoveries for different concentrations of the target bacteria ranged from 80% to 94.8%, showing the applicability of this method for the detection of *E. coli* O157:H7 in the milk. Notably, the assay’s versatility for other targets can be achieved through antibody modification. In another study, Ye et al. [60] employed a sandwich immunoassay strategy to detect *Cronobacter sakazakii* (*C. sakazakii*). This involved the construction of two types of nanocomplexes: silicon dioxide nanoparticles (SiNPs) encapsulating glucose oxidase and immunoglobulin G (IgG) for target recognition (SiNP-GOX-IgG), and magnetic nanoparticles (MNPs) functionalized with antibodies (MNP-IgG) (Figure 2B). When the target bacteria were present, a sandwich immunoassay complex of SiNPs-GOX-IgG/*C. sakazakii*/MNP-IgG formed, allowing the enzymatic hydrolysis of glucose by glucose oxidase within the complex, generating quantifiable PGM signals. And a linear relationship between the decrease in glucose concentration and the logarithm of *C. sakazakii* concentration was obtained. This method exhibited a sensitivity of 4.2 × 10^1^ CFU/mL and a linear range from 9.0 × 10^2^ to 9.0 × 10^7^ CFU/mL, providing high sensitivity. The entire incubation, separation, and detection process could be completed within 60 min, offering a rapid and simple approach for *C. sakazakii* detection. Luo et al. [61] also established an electrochemical quantification method for Salmonella detection by monitoring glucose consumption with a PGM platform based on the classic sandwich immunoassay (Figure 2C). The approach involved capturing and enriching target bacteria using antibody-functionalized magnetic nanoparticles (IgG-MNPs), followed by the conjugation of multiple monoclonal antibodies and glucose oxidase onto amino-functionalized silicon dioxide nanoparticles (IgG-SiNPs-GOx) as signal labels. In the presence of the target bacteria, a sandwich complex of IgG-MNPs/*Salmonella*/IgG-SiNPs-GOx formed, and glucose oxidase catalyzed the hydrolysis of glucose within the system, generating quantifiable signals through PGMs. Under optimal conditions, the limit of detection was 7.2 × 10^1^ CFU/mL, with a dynamic range ranging from 1.27 × 10^2^ to 1.27 × 10^5^ CFU/mL. This method presents an efficient and convenient approach for *Salmonella* detection.

### 3.2. Detection of Mycotoxins

Mycotoxins represent a diverse class of secondary metabolites produced by various fungi that thrive under conditions of high temperature and humidity, often contaminating food and a range of agricultural commodities including grains, nuts, spices, and coffee, among others [76]. Notably, mycotoxins such as aflatoxin B1 (AFB1), aflatoxin B2 (AFB2), aflatoxin G1, aflatoxin G2, ochratoxin A (OTA), trichothecenes, vomitoxin (DON), zearalenone (ZEN), and fumonisin B1 (FB1), produced by species including Aspergillus and Fusarium, have raised significant concerns [77]. Beyond their carcinogenic and mutagenic potential, these toxins are associated with adverse health effects on the kidneys, liver, hematopoietic system, immune system, and reproductive system. Hence, the development of rapid sensing platforms for mycotoxin detection in food is of paramount importance.

Exploring signal amplification strategies combined with PGMs is an essential direction in food safety detection [78]. It has found extensive application in the detection of mycotoxins. Yang et al. [62] employed aptamers as recognition elements, which are more easily modified than antibodies, and implemented signal amplification using a “DNA walker” to achieve rapid, cost-effective, and stable AFB1 detection (Figure 3A). Under optimized conditions, this method achieved a high sensitivity with a detection limit of 10 pm, and the recovery values of AFB1 content in moldy bread obtained were in the range of 98.5–103% with a relative standard deviation (RSDs) of 0.53–0.86% using the developed method. As depicted in Figure 3B, Tang et al. [63] employed a conventional sandwich immunoassay to capture AFB1, using liposome-encapsulated glucose as a signal amplification label. The liposome undergoes rapid hydrolysis upon exposure to PBST, facilitating quantitative glucose measurement based on PGMs. Under optimized conditions, the detection system exhibited a sensitivity as low as 0.6 pg/mL, significantly surpassing commercial ELISA kits and demonstrating excellent specificity for complex food matrices. And the recoveries were between 94% and 112%, revealing that the immunoassay could be employed for the detection of target AFB1 in complex systems. Gu et al. [64] and Qiu et al. [65] employed competitive PGM sensors for OTA detection in food (Figure 3C,D). In this strategy, a switchable OTA aptamer was immobilized onto magnetic beads, and competitive DNA carrying enzymatic labels was captured via base pairing. In the presence of the target, enzymatically labeled competitive DNA was released, resulting in the hydrolysis of sucrose into glucose by glucose oxidase and subsequent PGM-based quantification. The method yielded OTA detection limits of 3.31 μg/L and 72 pg/mL, respectively, offering a convenient and effective approach for OTA detection. And the average recoveries of Qiu et al.’s sensing system are in the range of 86–103.5%, while the RSDs are from 3.28% to 6.82%, showing the potential utility of the presented method for OTA detection in real samples.

Zhang et al. [66] designed a PGM sensing platform that integrated target recognition by aptamers and signal amplification by DNAzymes for OTA detection in food (Figure 3E). In this approach, a biotinylated aptamer for the target and a biotinylated substrate chain were conjugated onto magnetic beads, with a DNAzyme chain hybridized to the aptamer to inhibit substrate chain cleavage. In the presence of the target, the aptamer’s binding with specificity released the DNAzyme chain, enabling its hybridization with the substrate chain. In the presence of metal ions, substrate chain hydrolysis released free glucose, which was subsequently detectable via PGMs. The method achieved a LOD of 0.88 pg/mL for OTA, displaying robust specificity. And the recoveries obtained by the proposed method in this study ranged from 93.1% to 108.3%. Alterations in aptamer probes and DNAzyme chains facilitate broader utility for general small molecule analysis.

For the detection of patulin, Nie et al. [67] introduced a novel chemically linked strategy employing liposome-encapsulated glucose for signal amplification in PGM-based sensors. (Figure 3F). Additionally, this method innovatively employed thiol groups (-SH) as effective recognition moieties for patulin, synthesizing multifunctional thiol-modified liposomes (G-LIP-SH) as recognition elements for efficient patulin capture. Interaction with NH2-Au@Fe3O4 nanoparticles facilitated efficient and selective separation of G-LIP-SH not bound to patulin. Subsequent hydrolysis released encapsulated glucose, resulting in quantitative PGM signals inversely proportional to patulin concentration. The method achieved a detection limit of 0.05 ng/mL, and the recoveries were in the range of 90.6–98.9% with RSDs of 1.3–4.8%, exhibiting high specificity, repeatability, and accuracy, presenting an avenue for the development of rapid, portable, user-friendly, and cost-effective analytical methods.

### 3.3. Detection of Pesticide and Veterinary Drug Residues

In the current stage, the presence of a large number of toxic substances such as pesticides and veterinary drugs in food, along with issues of residues and contamination, has significantly impacted the quality and safety of food, posing serious threats to both the ecological environment and human lives [79]. Consequently, the development of expeditious detection methodologies for assessing the presence of agricultural and veterinary drug residues in food has emerged as a matter of the utmost necessity.

Kwon et al. [68] ingeniously employed the antibacterial activity of enrofloxacin to impede the glucose metabolism of *E. coli*, creating a rapid PGM sensing platform for enrofloxacin detection (Figure 4A). In the absence of enrofloxacin interference, glucose consumption by *E. coli* proceeds normally, resulting in decreased glucose levels. When enrofloxacin is present, glucose cannot be metabolized, leading to elevated glucose levels. This method achieves detection limits as low as 5 pg/mL in both water and milk matrices, well below the maximum residue limit of 100 ng/mL for enrofloxacin specified by GB 31650-2019 [80]. It demonstrates excellent selectivity, offering a convenient and effective approach for enrofloxacin detection. Li et al. [69] developed a PGMs sensing platform to detect ampicillin in milk (Figure 4B). This approach initially involves covalent coupling of magnetic beads (MBs) with ampicillin. In the absence of ampicillin in the sample, biotinylated aptamers completely bind to ampicillin-modified MBs. When ampicillin is present, competition for binding with biotinylated aptamers occurs, leading to the partial binding of biotinylated aptamers to ampicillin-modified MBs. Subsequently, less streptavidin is captured by the MBs, which in turn results in fewer biotinylated invertases being attached. After magnetic separation, the attached invertases are utilized to hydrolyze sucrose into glucose in the system. This establishes a correlation between ampicillin concentration in the sample and PGM-measured glucose. The method achieved a detection limit of 2.5 × 10^−10^ mol/L and a dynamic range of 2.5 × 10^−10^~1.0 × 10^−7^ mol/L, and the recoveries of spiked milk samples were between 83.57% and 126.49%, indicating that the proposed method might hold great potential for real sample detection. For ensuring consumer safety, as depicted in Figure 4C, Gao et al. [70] developed an immune-analytical method using PGMs for the rapid detection of norfloxacin in animal-derived food. Initially, AuNPs were employed as bridges to link invertase with anti-NOR monoclonal antibodies, serving as signal probes. NOR-BSA-functionalized MBs were used as sensing probes, competing with NOR to bind with the signal probes. Based on a competitive immune analytical format, after magnetic separation, the invertase within the complex converted sucrose into glucose, utilizing PGMs as signal transducers. This achieved a straightforward NOR detection method. The detection limit for NOR in standard solution was 0.5 ng/mL, with a broad linear range of 0.5~500 ng/mL, demonstrating good specificity. This immune PGM detection method provides an alternative for convenient and sensitive NOR detection in animal-derived food.

### 3.4. Detection of Heavy Metal Ions

Since the occurrence of environmental disasters such as “Minamata Disease” and “Itai-Itai Disease”, heavy metals have garnered sustained and widespread attention as significant environmental pollutants. These metals persist in the environment and can accumulate in organisms through the food chain, posing serious health risks to humans and animals [81]. Among them, cadmium has been established as toxic to kidney function and is also linked to osteoporosis. Its potential hazards include cardiovascular diseases, cancer, and immune system abnormalities [82]. Mercury is a highly toxic heavy metal, with a pronounced impact on the central nervous system, especially in fetuses and infants. Mercury poisoning can result in cognitive impairments, neurobehavioral abnormalities, and immune system dysregulation [83]. Their toxic effects are often insidious, making rapid detection of heavy metals in food crucial.

In the detection of heavy metal ions using PGM-based biosensors, DNAzymes are predominantly employed as recognition elements. For instance, Zhang et al. [71] developed a simple and cost-effective DNA sensing platform for the highly sensitive detection of Pb^2+^ in environmental samples, based on a PGM detection scheme utilizing DNAzyme-modified microplates (Figure 5A). In this method, DNAzymes were immobilized on streptavidin-modified microplates, and AuNPs labeled with single-stranded DNA and invertase (Enz-AuNPs-DNA) served as signal transduction labels. When the target Pb^2+^ was present, the substrate strand of immobilized DNAzyme was cleaved by Pb^2+^, generating new single-stranded DNA in the microplate wells. This new DNA could hybridize with the single-stranded DNA on Enz-AuNPs-DNA, carrying the invertase to convert sucrose to glucose, which was monitored by PGMs. The method achieved a detection limit as low as 1.0 pM for Pb^2+^ under optimal conditions, and the RSDs in different concentrations ranged from 3.4% to 12.1%, exhibiting good reproducibility and high selectivity, rendering it a convenient and effective Pb^2+^ detection method.

In contrast to the DNAzymes with substrate cleavage activity mentioned earlier, Zeng et al. [72] employed a signal amplification strategy based on the specific recognition of DNA3 and Exo III to quantitatively detect Cd^2+^ in food samples. As shown in Figure 5B, in the presence of the target Cd^2+^, the adaptor strand specifically recognized Cd^2+^, and the 3’ to 5’ exonuclease activity of DNA exonuclease III (Exo III) selectively digested DNA2, triggering a cyclic signal amplification process. Through consecutive hybridization and cleavage reactions, a large amount of single-stranded DNA was released from the magnetic beads. The invertase-conjugated DNA complementary to the released single-stranded DNA was introduced into the sensing system, and after magnetic separation, the invertase complex hydrolyzed sucrose to glucose, converting Cd^2+^ concentration to glucose amount, which was quantified directly by PGMs. The synergistic signal amplification of Exo III and the enzyme significantly enhanced the sensitivity of Cd^2+^ analysis, achieving a detection limit of 5 pM. The method offered portability, cost-effectiveness, wide availability, and user-friendliness, making it a promising tool for the routine quantitative detection of Cd^2+^.

### 3.5. Detection of Illegal Additives

Illegal additives refer to substances prohibited by national regulations for use in food products. These substances often possess strong carcinogenic and teratogenic properties, posing significant risks to consumer health [84]. For instance, cocaine is a central nervous system stimulant renowned for its capacity to elevate dopamine levels and efficiently impede the reuptake of neurotransmitters at synapses, which ranks as the second most used illegal substance in both Europe and the United States [85]. Cocaine abuse is associated with detrimental effects such as anxiety, paranoia, mood disorders, organ damage, and the hallmark of violent behavior. Clenbuterol (CLB) is a growth promoter which is completely banned in some countries, an example being China’s prohibition of CLB in animal feed as early as 1997. Despite repeated bans, its illegal use for inducing weight gain and a higher ratio of muscle to fat, particularly in the pig farming industry, persists [86]. Establishing rapid and on-site detection methods for illegal additives in food products is, therefore, imperative.

For instance, as illustrated in Figure 6A, Fang et al. [73] employed a competitive enzyme-linked immunosorbent assay (ELISA) combined with PGMs to detect CLB. This approach involved the synthesis of two types of nanocomposites: CLB-bovine-serum-albumin-functionalized magnetic beads (CLB-BSA MB) and AuNPs–invertase–antibody conjugates. The presence of CLB competed with CLB-BSA for binding to antibodies, inhibiting the further assembly of CLB-BSA MB with AuNPs–invertase–antibody conjugates. Consequently, the collected invertase was reduced, leading to a decline in PGM signal. The logarithmic concentration of CLB exhibited a correlation with the signal ratio of PGMs before and after CLB addition. The linear dynamic range for CLB detection ranged from 0.1 to 100 ng/mL, with a detection limit of 0.1 ng/mL, meeting the maximum residue limit established by the Food and Agriculture Organization of the United Nations for practical testing. And the recoveries of spiked pork and liver samples were between 83.4% and 96%, indicating that the developed method can be applied to CLB residue detection in real samples. This method demonstrated robust specificity and universality for other non-glucose substances through antibody replacement. Recently, Tang and colleagues [74] devised a simple and portable PGM-based biosensing platform for quantitative cocaine detection based on a TiO_2_ nanotube array (TiNTA)-mediated glucose release system (Figure 6B). Initially, single-stranded DNA1 and DNA2 were covalently attached to TiNTA and AuNPs, respectively. In the presence of cocaine’s aptamer and glucose, DNA1-TiNTA immobilized on TiNTA and DNA2-AuNPs labeled with DNA2 both hybridized with adjacent regions of cocaine’s aptamer, encapsulating glucose molecules within the nanotubes. Upon the introduction of cocaine, the aptamer’s specific reaction caused the dissociation of DNA1-TiNTA and DNA2-AuNPs, releasing glucose molecules from the nanotubes for PGM-based quantification. Experimental results demonstrated an increase in PGM signal with rising cocaine concentrations, exhibiting a broad linear range of 10 to 600 nM and a detection limit of 5.2 nM. And the recoveries were between 89.6% and 117%, revealing that the method could be utilized for the quantitative detection of cocaine in complex systems. This method displayed favorable specificity, repeatability, and stability, offering a reference for the rapid, simple, cost-effective, and user-friendly detection of cocaine.

## 4. Prospects of PGM-Based Biosensors in Food Safety Hazard Detection

From the first detection of non-glucose substances to their current widespread application and development, PGM-based biosensors have traversed more than a decade. In comparison to traditional instruments employed for food safety hazard detection, PGM biosensors offer heightened portability, rapidity, and cost-effectiveness, effectively catering to the demands of on-site testing. In contrast to other biosensors such as colorimetric sensors, PGM-based biosensors typically exhibit more stable electrochemical signals, demonstrating good recovery rates and low RSDs. These characteristics signify the potential of this method for the quantitative detection of food safety hazards in complex systems. Contrasting with other electrochemical biosensors, such as smartphones, due to potential variations among different smartphone models, PGMs, as a highly commercialized and straightforward reading platform, offer a more stable and intuitive quantitative output signal. Within the domain of rapid food safety detection, these biosensors exhibit remarkable advantages and applicability.

In this paper, we provide a comprehensive overview of the mechanism underlying the construction of PGM-based biosensors, which consists of three fundamental components: recognition, signal transduction, and signal output. This paper also detailedly enumerates available recognition and signal transduction elements, and their modes of integration. Then, a multitude of instances is examined to present the latest advancements in the application of PGMs in food safety hazard detection, including targets such as pathogenic bacteria, mycotoxins, agricultural and veterinary drug residues, heavy metal ions, and illegal additives.

While there has been some progress, PGM-based biosensors still face numerous challenges when applied to food safety hazard detection. As a continually evolving and refining detection modality, the future directions for the development of PGM-based biosensors in food safety hazard detection are described as follows:(1)The development of additional recognition mechanisms is necessary. Antibodies, aptamers, and DNAzymes have been widely employed as direct recognition elements in the construction of PGM-based sensors. Given the diverse range of food safety hazards, it is essential to develop alternative, more efficient, and applicable recognition mechanisms to broaden their applicability, such as phages’ highly specific recognition [87] and click chemistry [88].(2)The coupling strategy between recognition elements and signal transduction components requires further optimization. The binding of oligonucleotides, antibodies, and other recognition elements to enzymes often leads to reduced enzymatic efficiency and escalated experimental costs. Efforts have been directed toward the design and development of antibody-enzyme fusion proteins to ascertain their potential universality and commercial viability [52].(3)The fabrication of LFA-PGM sensing platforms faces certain constraints. Firstly, the current setup still necessitates manual intervention in the cutting of LFA detection lines and their integration with PGMs. Seamlessly bridging the gap between LFA and PGMs warrants careful consideration. Secondly, the potential loss of enzymes within the paper-based chromatography of LFA could potentially compromise sensitivity.(4)Tedious pre-processing steps impact detection time. Although MNPs find extensive application in PGM biosensors, the intricate magnetic separation and washing procedures pose challenges to rapid on-site testing. DNA nanoflowers with recognition and separation capabilities [89] hold promise as alternatives to MNPs.(5)While PGMs have achieved significant commercialization, research into PGM-based biosensors remains primarily within the laboratory domain. Currently, there is a dearth of matured and enhanced PGM detection equipment suitable for practical commercial applications. Factors such as cost, storage requirements, shelf life, and strategies for mitigating interference from endogenous sugar sources are critical determinants in facilitating the commercialization of PGM-based biosensors [12].

## Figures and Tables

**Figure 1 foods-12-03947-f001:**
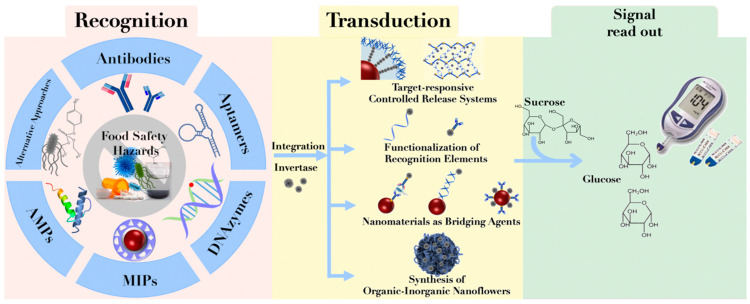
Construction scheme of PGM-based biosensors.

**Figure 2 foods-12-03947-f002:**
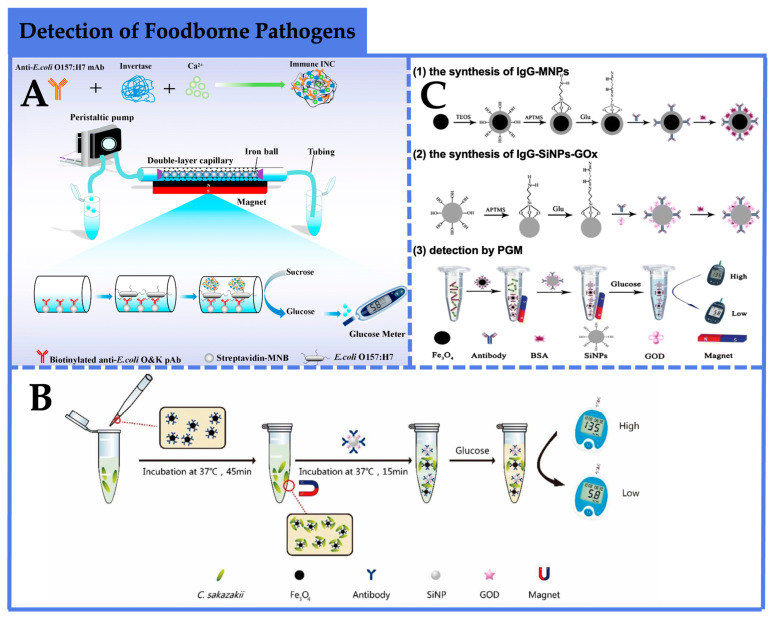
Application of PGM-based biosensors in the detection of foodborne pathogens. (**A**) A sensitive biosensor using double-layer capillary-based immunomagnetic separation and invertase-nanocluster-based signal amplification for the rapid detection of foodborne pathogens [59]. Copyright 2017, Elsevier. (**B**) A sandwich immunoassay strategy to detect *Cronobacter sakazakii* (*C. sakazakii*) [60]. Copyright 2017, Royal Society of Chemistry. (**C**) An electrochemical quantification method for *Salmonella* detection using the PGM platform based on the classic sandwich immunoassay [61]. Copyright 2017, Springer Nature.

**Figure 3 foods-12-03947-f003:**
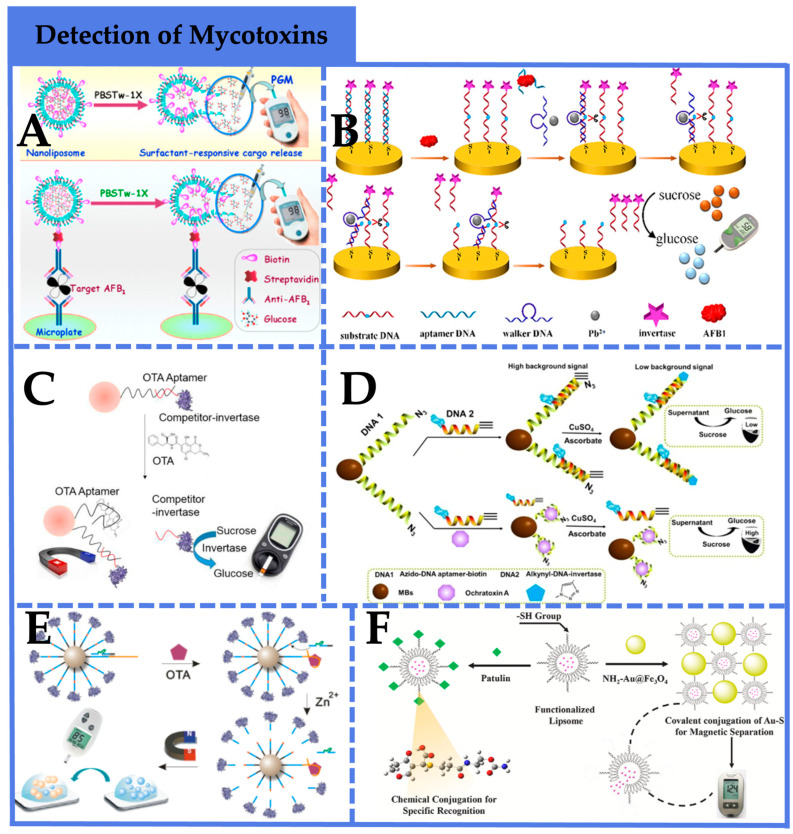
Application of PGM-based biosensors in the detection of mycotoxins. (**A**) A PGM-based biosensor-implemented signal amplification using a “DNA walker” to achieve rapid, cost-effective, and stable AFB1 detection [62]. Copyright 2018, American Chemical Society. (**B**) A conventional sandwich immunoassay to capture AFB1, using liposome-encapsulated glucose as a signal amplification label [63]. Copyright 2016, Elsevier. (**C**) A competitive PGM sensor for OTA detection in red wine [64]. Copyright 2016, The Royal Society of Chemistry. (**D**) A competitive PGM sensor for OTA detection in food [65]. Copyright 2019, Elsevier. (**E**) A PGM sensing platform that integrated target recognition by aptamers and signal amplification by DNAzymes for OTA detection in food [66]. Copyright 2021, American Chemical Society. (**F**) A novel chemically linked strategy employing liposome-encapsulated glucose for signal amplification in PGM-based sensors [67]. Copyright 2020, Elsevier.

**Figure 4 foods-12-03947-f004:**
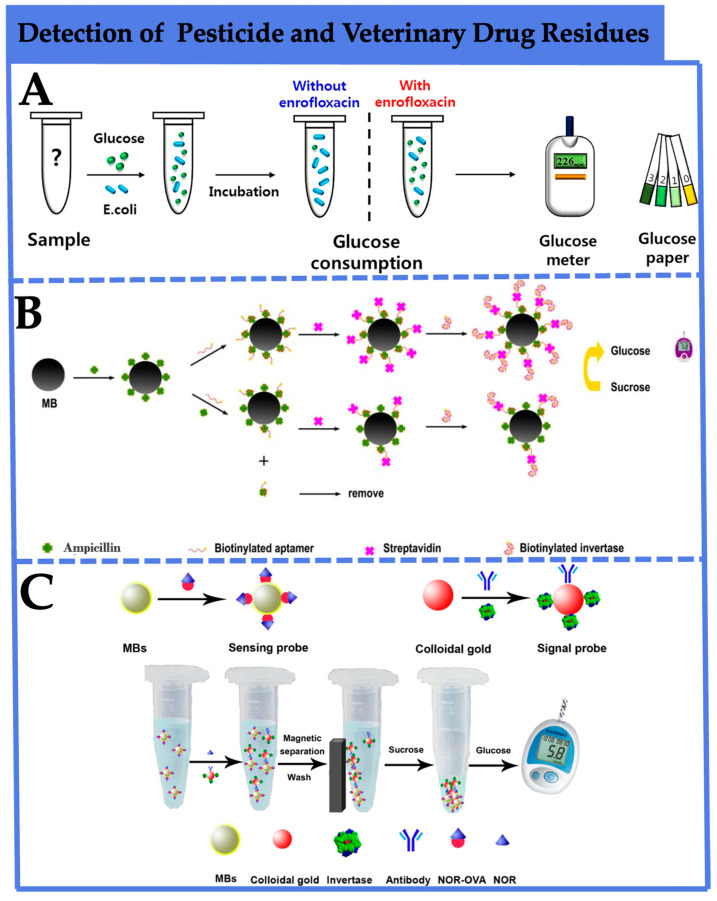
Application of PGM-based biosensors in the detection of agricultural and veterinary drug residues. (**A**) A rapid PGM sensing platform for enrofloxacin detection [68]. Copyright 2018, Elsevier. (**B**) A PGM sensing platform to detect ampicillin in milk [69]. Copyright 2020, The Royal Society of Chemistry. (**C**) An immune-analytical method using PGMs for the rapid detection of norfloxacin in animal-derived food [70]. Copyright 2021, Springer Nature.

**Figure 5 foods-12-03947-f005:**
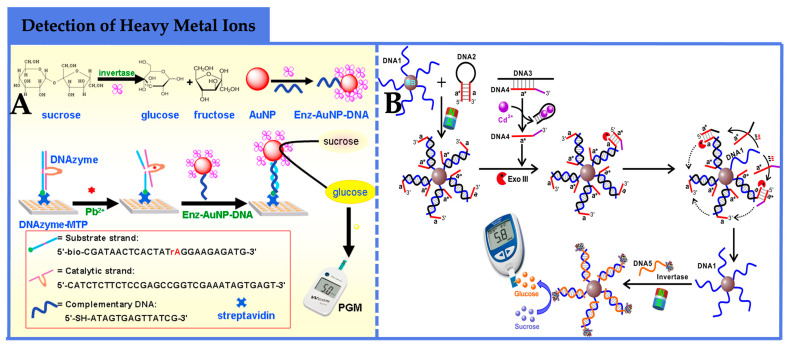
Application of PGM-based biosensors in the detection of heavy metal ions. (**A**) A simple and cost-effective DNA sensing platform for the highly sensitive detection of Pb^2+^ in environmental samples, based on a PGM detection scheme utilizing DNAzyme-modified microplates [71]. Copyright 2015, Elsevier. (**B**) A signal amplification strategy based on the specific recognition of DNA3 and Exo III to quantitatively detect Cd^2+^ in food samples [72]. Copyright 2019, Elsevier.

**Figure 6 foods-12-03947-f006:**
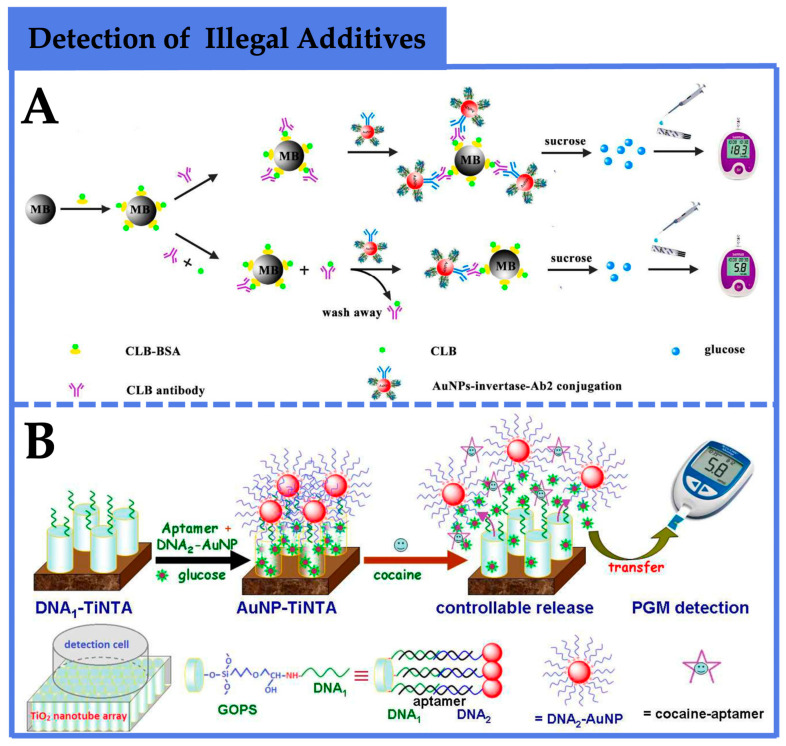
Application of PGM-based biosensors in the detection of illegal additives. (**A**) A competitive enzyme-linked immunosorbent assay (ELISA) combined with PGMs to detect CLB [73]. Copyright 2017, The Royal Society of Chemistry. (**B**) A simple and portable PGM-based biosensing platform for quantitative cocaine detection based on a TiO_2_ nanotube array (TiNTA)-mediated glucose release system [74]. Copyright 2017, The Royal Society of Chemistry.

**Table 1 foods-12-03947-t001:** Recent applications of PGM-based biosensors in food hazard detection.

Target Category	Target	Bioreceptor	Signal Transduction Elements	Signal Transduction Strategies	Detection Samples	LOD	References
Foodborne pathogens	*E. coli* O157:H7	Antibody	Invertase	Nanomaterials as Bridging Agents	Physiological saline, milk	6.2 × 10^4^ CFU/mL	[17]
		Antibody	Invertase	Organic–Inorganic Nanoflowers	Milk	79 CFU/mL	[59]
		Con A	Invertase	Organic–Inorganic Nanoflowers	Milk	10^1^ CFU/mL	[58]
		AMPs	Invertase	Organic–Inorganic Nanoflowers	Milk	10 CFU/mL	[38]
	*C. sakazakii*	Antibody	Invertase	Nanomaterials as Bridging Agents	Milk powder	4.2 × 10^1^ CFU/mL	[60]
	*E. coli*	Bacterial glycolysis	Glucose	—	Tap water	2 × 10^6^ CFU/100 µL	[39]
	*Salmonella*	Antibody	Invertase	Nanomaterials as Bridging Agents	milk	10 CFU/mL	[56]
		Antibody	Glucose oxidase	Nanomaterials as Bridging Agents	Meat broth medium	7.2 × 10^1^ CFU/mL	[61]
	*Staphylococcus aureus*	Aptamer	Invertase	Nanomaterials as Bridging Agents	Peach juice, milk, water	2 CFU/mL	[23]
Mycotoxins	AFB1	Aptamer	Glucose	Controlled Release Systems	Buffer	0.02 ng/mL	[49]
		Aptamer	Invertase	Functionalization of Recognition Elements	Moldy bread	10 pm	[62]
		Antibody	Glucose	Controlled Release Systems	Buffer	0.6 pg/mL	[63]
	OTA	Aptamer	Invertase	Functionalization of Recognition Elements	Buffer, red wine	3.31 μg/L, 3.66 μg/L	[64]
		Aptamer	Invertase	Nanomaterials as Bridging Agents	Feed	72 pg/mL	[65]
		Aptamer	Invertase	Nanomaterials as Bridging Agents	Wine	0.88 pg/mL	[66]
	*patulin*	-SH	Glucose	Controlled Release Systems	Grape juice	0.05 ng/mL	[67]
Agricultural and veterinary drug residues	CAP	MIPs	Invertase	Nanomaterials as Bridging Agents	Fish and pork	0.16 ng/mL	[34]
	enrofloxacin	*E. coli*	Glucose	—	Water and milk	5 ng/mL	[68]
	paraoxon	Acetylcholinesterase	[Fe(CN)6]^3−^	Thiocholine byproduct	Apple and cucumber	5 µg/mL	[40]
	ampicillin	Aptamer	Invertase	Nanomaterials as Bridging Agents	milk	2.5 × 10^−10^ mol/L	[69]
	norfloxacin	Antibody	Invertase	Nanomaterials as Bridging Agents	Milk, chicken, pork, shrimp	0.5 ng/mL	[70]
Heavy metal ions	Pb^2+^	DNAzymes	Glucose	Controlled Release Systems	Drinking water	1 pM	[27]
		DNAzymes	Invertase	Functionalization of Recognition Elements	Wastewater, drinking water	1.0 pM	[71]
	Cd^2+^	Aptamer	Invertase	Functionalization of Recognition Elements	Lake water, pond water	5 pM	[72]
Illegal additives	CLB	Antibody	Invertase	Nanomaterials as Bridging Agents	Pork, liver	0.1 ng/mL	[73]
	melamine	Aptamer	Invertase	Nanomaterials as Bridging Agents	Buffer, 80% full-fat milk	0.33 µM, 0.53 µM	[22]
	cocaine	Aptamer	Glucoamylase	Controlled Release Systems	—	3.8 μM	[50]
		Aptamer	Glucose	Controlled Release Systems	—	5.2 nM	[74]

Note: The use of ‘—’ denotes information that has not been specified or provided. (LOD: limit of detection; Con A: concanavalin A; AMPs: antimicrobial peptides; AFB1: aflatoxin B1; OTA: ochratoxin A; -SH: thiol groups; CAP: chloramphenicol; MIPs: molecularly imprinted polymers.).

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
