# Peer review of "Recent Advances in Personal Glucose Meter-Based Biosensors for Food Safety Hazard Detection"

_foods, 2023, doi:10.3390/foods12213947_

Round 1

Reviewer 1 Report

The authors present a well-structured and engaging review of the use of PGM as a strategy for building portable, robust transduction, and readout devices.

This reviewer believes that the manuscript can be considered for publication in this journal with just a few minor corrections and suggestions:

1) I would suggest to the authors that they include a column in Table 1 explaining the type of signal transduction strategy used in each case (e.g., controlled release systems, labeling of recognition elements, nanomaterials as bridging agents, organic-inorganic nanoclusters) or directly modify the column "signal transduction elements" to reflect this information.

2) Please, be mindful of the established terminology in the field. In Table 1, "Identification elements" should be changed to "Bioreceptor" or "biological receptor." Likewise, it would be better to use "transduce" instead of "transmute" or "translate." The authors have used the words "transmute," "translate," and "transduce," and they should choose one term for clarity for the readers (ideally use "transduce").

3) On page 6, line 247, "intergration stratrgies" should be corrected to "integration strategies."

4) Please add a table footnote explaining the acronyms used in the table.

5) On page 5, line 193, the AMP used was not AMP I and AMP P1 but rather magainins I and cecropin P1.

6) The entries in Table 1 with "The inherent characteristics of the target" should be changed:

- In reference 36, I would suggest "bacterial glycolysis." Also, I would say that this reference can hardly be considered a biosensor. First, no biorecognition element is used, just the consumption of glucose by E. coli, more like an assay for the detection of bacteria. Glycolysis is not specific to E. coli.

- In reference 64, I would suggest replacing "The inherent characteristics of the target" with "E. coli." The researchers used E. coli as a bioreceptor, where enrofloxacin obstructs bacterial metabolism and reduces glucose consumption in proportion to the E. coli concentration.

- In reference 37, I would suggest replacing "The inherent characteristics of the target" with "acetylcholinesterase." The detection of organophosphorus compounds is based on the principle that these compounds reduce the yield of the enzyme's hydrolysis of acetylthiocholine chloride. The signal transduction mechanism of the same reference is ferro/ferricyanide, but this is true for many second-generation glucose meters (i.e. all PGM). Therefore, ferro/ferricyanide is not an element added into the device by the researchers, but already present in the PGM. I would suggest changing the entry in the able to "thiocholine byproduct," which is the active element that diffuses into the PGM and drives the redox reaction with the ferro/ferricyanide."

Reviewer 2 Report

The review presents a useful analysis of current activity in the use of portable glucose meters as registering tools for different assays and sensors. This direction of developments growths actively, and analysis of its state-of-the art and perspectives is in demands, first of all – for efficient point-of-care diagnostics. The prepared manuscript will be useful for readers of the Biosensors journal, but needs improvements:

1. PGMs as Personal Glucose Meters is not the abbreviation of definitely and general knowledge (Platinum Group Metals? Pragmatic General Multicast?). So it should not be used in the title.

2. References 2-5 present specific developments for concrete targets and so are not good confirmations concerning overall state-of-the-art at lines 35-37. Please give some recent reviews/monographs here.

3. Please indicate in the Introduction refs. 10, 11 namely as predecessors reviewing the same thematic field and clarify specific features of the new review.

4. It will be useful to consider separation of the described techniques with glucose detection as «on» and «off» ones, i.e. with generation (increasing) and with transformation (decreasing) of glucose molecules in the course of specific analytical reactions.

5. Lines 107-108. Antibodies were used for analytical purposes in such techniques as immunoprecipitation etc. much earlier than the development of radioimmunoassay.

6. Lines 112-118. The authors cite the work of Dou et al. [16] (probably Huang et al. – see line 643) published in 2018 and state that the following integration of PGMs and test strips will be an avenue for further optimization. However, this integration was continued for food-related analytes (see DOI 10.1007/s00217-021-03825-8) and non-food analytes (see DOI 10.3390/s21020660). So the description in the Section 2.1.1 needs correction.

7. Lines 154-156 state universality of DNAzymes use for the detection of different heavy metal ions. However, this statement is not grounded. Have any other glucose-releasing DNAzymes whose activity is selectively regulated by other heavy metal ions been described?

8. Lines 158-162. K. Mosbach and co-authors have proposed MIPs before the cited work [27]. Se, for example, DOI 10.1016/0003-2697(89)90029-8 (1989), DOI 10.1016/S0021-9673(01)89434-6 (1990).

9. Lines 173-177. The description of work [31] needs in indication of the use of glucose meter and specification of process(es) causing change of glucose content.

10. The relations between the processes described in lines 208-217 and changes of glucose levels in the reaction media need additional clarification.

11. «Signal transduction elements» is a very general term. Therefore, please give a specification of which of the molecules responsible for different types of transduction can provide the signal recorded by the glucometer?

12. Possible arrangements of (i) carriers/solutions for glucose generation/transformation and (ii) sensing parts of glucose meters should be clarified.

13. Multiple risks of environmental and food pollutions caused by heavy metals are well-known and are not limited by specific examples given at lines 498-489. The corresponding comments should be modified.

14. Short legends to Figures 2-6 should be accomplished by addressing to text of the review giving detailed explanation of their elements. I think also that references in the figures' legends should be associated with parts of the figures ((a) … [59], (b) … [60], (c) … [61] etc.) for better indication of copyright issues.

15. I doubt that cocaine should be considered in relation to food safety hazards (see lines 543-556). On another hand, clenbuterol can be better specified as growth promoter instead on unclear indication as illegal additive. So the Section 3.5 needs transformation.

16. At whole, Section 3 presents few examples for each group of toxicants based on different assay principles, and the choice of these principles is not strongly related with toxicants' group; i.e. approaches for mycotoxins are potentially applicable for veterinary drugs, and vice versa and so on. As a result, the overall variety of biosensors retain non-characterized. I strongly recommend to consider restructuring of this Section with list of basic biosensoric formats regardless target analytes.

17. The integral comments in the Section 4 are very common and do not cover all aspects of PGMs use. The following issues will be important for understanding practical potential and perspectives of PGMs-based assays for food safety control:

- How accurate are toxicant measurements using glucometers? Please consider typical RSD values.

- What are the advantages and disadvantages of glucometers as recording tools compared to portable optical detectors (including smartphones), which are also actively used for recording the results of POC tests?

18. Some references seem incomplete – [6], [24], [30], [61], [74], [76]. Refs. [70] and [71] are the same one.

Reviewer 3 Report

see attached file

Minor editing of English language required

Round 2

Reviewer 2 Report

The manuscript has been successfully revised and now may be published